# MolGene-E: Inverse Molecular Design to Modulate Single Cell Transcriptomics

## Abstract

Designing drugs that can restore a diseased cell to its healthy state is an emerging approach in systems pharmacology to address medical needs that conventional target-based drug discovery paradigms have failed to meet. Single-cell transcriptomics can comprehensively map the differences between diseased and healthy cellular states, making it a valuable technique for systems pharmacology. However, single-cell omics data is noisy, heterogeneous, scarce, and high-dimensional. As a result, no machine learning methods currently exist to use single-cell omics data to design new drug molecules. We have developed a new deep generative framework named MolGene-E that can tackle this challenge. MolGene-E combines two novel models: 1) a cross-modal model that can harmonize and denoise chemical-perturbed bulk and single-cell transcriptomics data, and 2) a contrastive learning-based generative model that can generate new molecules based on the transcriptomics data. MolGene-E consistently outperforms baseline methods in generating high-quality, hit-like molecules from gene expression profiles obtained from single-cell datasets and gene expressions induced by knocking out targets using CRISPR. This superior performance is demonstrated across diverse *de novo* molecule generation metrics, which makes MolGene-E a potentially powerful new tool for drug discovery.

## 1 Introduction

Capitalizing on the success of deep learning across various domains such as natural language, images, and videos, deep generative models have been extensively applied to the generation of small organic compounds targeting a specific disease gene for drug discovery Zeng et al. (2022). However, this one-drug-one-target paradigm has had limited success in tackling polygenic, multifactorial diseases. Due to the high costs, prolonged development timelines, and low success rates associated with target-based drug discovery, there has been a resurgence of interest in phenotypic drug discovery. As a matter of fact, approximately 90% of approved drugs have been discovered through a phenotype-driven approach Vincent et al. (2022). Therefore, phenotype-based molecular generation, also known as inverse molecule design, holds promise for the discovery of novel therapeutics aimed at addressing medical needs that conventional target-based drug discovery paradigms have failed to meet.

The effectiveness of phenotype-based drug discovery relies upon the careful selection of an appropriate phenotype readout. Chemical-induced transcriptomics has been embraced as a comprehensive systematic measurement for phenotype drug discovery. The transcriptomic change resulting from chemical exposure can function as a chemical signature for predicting drug responses as well as aid in the elucidation of drug targets and the inference of drug-modulated pathways. This approach has demonstrated successful applications in phenotype drug repurposing Salame et al. (2022) Pham et al. (2021). Several deep learning methods have been proposed to leverage chemical-induced bulk gene expression data for inverse molecule design. Notably, MolGAN Méndez-Lucio et al. (2020) generates molecules conditioning a generative adversarial network with bulk transcriptomics data. Although it shows promising results, GANs are susceptible to scalability, as we show that their performance drops significantly when trained on higher dimensional data. Furthermore, GANs have a black-box nature, and inferring the relation between the condition (gene expression) and generation (molecules) is quite cumbersome. Another recent work is the GxVAE Li & Yamanishi (2024), which employs two joint variational autoencoders (VAEs) to facilitate the extraction of latent gene

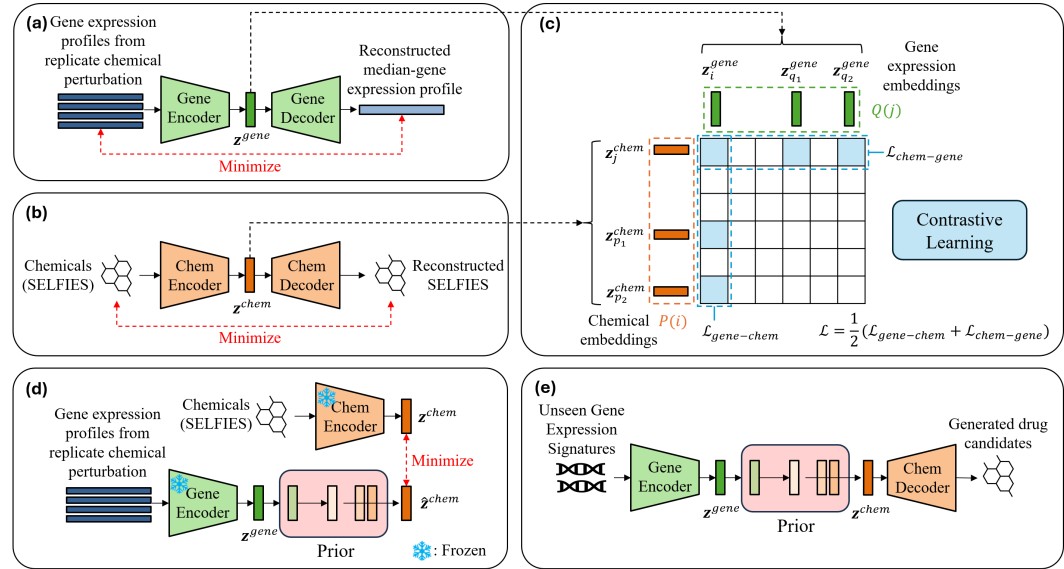

Figure 1: **(a)** A Variational Autoencoder (VAE) denoises the gene expression profiles corresponding to replicate chemical perturbation in a batch by reconstructing their median gene expression profiles. **(b)** We represent chemical structures via SELFIES and use a pretrained frozen VAE to extract the chemical embeddings. **(c)** The gene expression encoder is fine-tuned to align gene embeddings $z^{gene}$ to the chemical embeddings $z^{chem}$ via a contrastive learning module. A supervised objective $\mathcal{L}$ (Equation 3) is optimized to maximize the agreement between positive pairs while minimizing the similarity between negative pairs. **(d)** A prior model is trained to map the inferred $z^{gene}$ to the inferred $z^{chem}$. **(e)** Given unseen gene expression profiles, the inferred $z^{gene}$ are mapped to $z^{chem}$ via the pretrained prior model, and are further decoded using the SELFIES VAE's decoder to generate drug candidates.

expression features and using it as a condition to generate molecules using a second VAE. However, GxVAE has not been developed for single-cell transcriptomics data.

The abundance of single-cell omics data provides new opportunities for phenotype-based drug discovery. Single-cell transcriptomics data offer new insights into disease heterogeneity within and across species, illuminating the complexity of pathological processes. An effective therapy often needs to modulate disease etiology at the single-cell level Han et al. (2022). Furthermore, precise characterization of single-cell chemical transcriptomics is crucial to bridge translational gaps between disease models (e.g., organoids and animals) and human patients, a critical bottleneck in drug discovery Van de Sande et al. (2023). Nonetheless, there remains a scarcity of methods for leveraging single-cell transcriptomics data in inverse molecule design.

Compared with protein structures that exhibit a relatively clean nature, omics data is plagued by its high-dimensionality and susceptibility to noise, stemming from biological stochasticity and technical artifacts. These complexities pose hurdles for single-cell inverse molecule design, exacerbated by the limited availability of chemical-perturbed single-cell transcriptomics data. LINCS1000 Subramanian et al. (2017) serves as a comprehensive chemical transcriptomics database, profiling 19,811 chemicals across 77 cell lines. However, this database profiles only 978 landmark genes. Moreover, the gene expression data in LINCS1000 is obtained using a specific imaging technique, leading to significant distributional discrepancies from RNA-seq data. Due to these challenges, no methods exist for inverse molecule design based on single-cell omics data.

To address these challenges, we introduce MolGene-E, a deep learning framework for single-cell molecule generation. The key contributions of MolGene-E are twofold: First, we develop a domain adaptation model that is capable of harmonizing and denoising L1000toRNAseq and Sciplex-3 single-cell chemical transcriptomics data. Second, we design a generative algorithm that leverages

contrastive learning to align phenotypic representations to chemical representations, by integrating these components, MolGene-E facilitates the generation of novel molecules with specific phenotypic traits. Extensive evaluations demonstrate that MolGene-E achieves state-of-the-art performance, positioning it as a potentially powerful new tool for drug discovery.

## 2 RELATED WORK

To leverage chemical-induced bulk gene expression data for inverse molecule design, MolGAN Méndez-Lucio et al. (2020) exploited gene expression profiles to generate hit-like molecules. However, the conditioning network in MolGAN makes use of real and fake conditions, where conditions outside of the training batch are assumed to be fake, which could introduce potential inaccuracies due to the presence of multiple replicates with distinct gene expressions in the L1000 dataset. A more recent work, GxVAE Li & Yamanishi (2024), proposed using joint VAEs to extract latent features from gene expressions and using them as conditions for generating molecules utilizing a second VAE for encoding molecules. These approaches result in poorly structured latent space, consequently affecting the model's ability to generate chemically valid molecules. Additionally, these models do not address the incumbent challenge of leveraging highly sparse and out-of-distribution single-cell datasets to generate molecules. To address these challenges, we developed a denoising gene expression encoder, which reconstructs the median gene expression in the case of replicates chemical samples, essentially acting as a denoising gene expression autoencoder. This approach ensures a more meaningful and robust representation of gene expressions, contributing to a better-structured latent space and enhancing the generation of valid and diverse molecular structures.

## 3 METHODS

MolGene-E is a framework designed for the *de novo* generation of molecules with desired biological properties. As shown in Figure 1, the framework involves a five-step process that integrates diverse data representations and deep learning modules to align chemical and gene expression profile information effectively.

### 3.1 DENOISING VAE FOR GENE EXPRESSION PROFILES

In chemical-induced bulk gene expression data, multiple distinct gene expression profiles perturbed by replicate chemicals can exist. In order to manage and interpret the complex data from multiple replicates, MolGene-E employs a Variational Autoencoder (VAE). The VAE is trained with the objective of reconstructing the median gene expression profile from the gene expression profiles corresponding to replicate chemical perturbations in a batch (Figure 1a). This approach ensures that the VAE captures the most representative gene expression profile, smoothing out anomalies and focusing on the core response to chemical perturbations. This process enhances the reliability of the gene expression data used in further steps.

### 3.2 SELFIES VAE FOR CHEMICALS

For representing the chemical structures in perturbations, MolGene-E leverages SELFIES (Self-Referencing Embedded Strings) Krenn et al. (2020) due to its guaranteed 100% validity in contrast to using SMILES strings to represent molecules for molecule design Méndez-Lucio et al. (2020). These SELFIES strings are encoded using a VAE model pretrained on ZINC dataset Gao et al. (2022) (Figure 1b). The use of SELFIES allows for a comprehensive and error-resistant encoding of molecular structures, facilitating seamless integration with machine learning models.

### 3.3 ALIGNMENT OF GENE EXPRESSIONS AND CHEMICAL REPRESENTATIONS

The key innovation in MolGene-E lies in aligning the gene expression profiles with their corresponding chemical perturbations. This is achieved through a contrastive learning module (Figure 1c) trained with a supervised contrastive loss $\mathcal{L}$ (Equation 3) inspired by CLIP Radford et al. (2021) and SupCon loss Khosla et al. (2020). The objective of this module is to align the embeddings of phenotypes (gene expression profiles) with the embeddings of the SELFIES representations of the

chemicals that caused the perturbations. It is also specifically designed to deal with the existence of multiple distinct gene expression profiles perturbed by replicate chemicals in a batch. By doing so, MolGene-E ensures that the biological effects of chemicals are accurately reflected in their encoded representations.

To define the contrastive loss, we introduce $\mathcal{L}_{\text{gene-chem}}$ and $\mathcal{L}_{\text{chem-gene}}$. The former aligns each arbitrary anchor gene expression embedding $z_i^{gene}$ with an index $i$ to all corresponding replicate chemical perturbation embeddings $z_p^{chem}$ with indices $p \in P(i)$ in a batch:

$$\mathcal{L}_{\text{gene-chem}} = \sum_{i \in I} \frac{-1}{|P(i)|} \sum_{p \in P(i)} \log \frac{\exp\left(z_i^{gene} \cdot z_p^{chem}/\tau\right)}{\sum_{k \in I} \exp\left(z_i^{gene} \cdot z_k^{chem}/\tau\right)}, \tag{1}$$

while the latter aligns each arbitrary anchor chemical embedding $z_j^{chem}$ with an index $j$ to all corresponding perturbed gene expression embeddings $z_q^{gene}$ with indices $q \in Q(j)$ in a batch:

$$\mathcal{L}_{\text{chem-gene}} = \sum_{j \in I} \frac{-1}{|Q(j)|} \sum_{q \in Q(j)} \log \frac{\exp\left(z_j^{chem} \cdot z_q^{gene}/\tau\right)}{\sum_{k \in I} \exp\left(z_j^{chem} \cdot z_k^{gene}/\tau\right)}. \tag{2}$$

In the two equations above, $\tau$ denotes the temperature parameter controlling the sharpness of the similarity scores, $|P|$ denotes the cardinality of $P$, and $I$ is the set of all indices in the batch.

The final contrastive loss $\mathcal{L}$ is obtained by combining the two losses above:

$$\mathcal{L} = \frac{1}{2}\left(\mathcal{L}_{\text{gene-chem}} + \mathcal{L}_{\text{chem-gene}}\right), \tag{3}$$

The SELFIES chemical embeddings are used directly from the pre-trained chemical encoder model underscored in the previous section and its parameters are frozen while training.

### 3.4 MAPPING GENE EXPRESSIONS TO CHEMICAL EMBEDDINGS

To complete the alignment process, MolGene-E employs a Multi-Layer Perceptron (MLP)-based prior model (Figure 1d). This model is trained to map the embeddings of gene expression profiles to the embeddings of their corresponding chemical counterparts. The MLP-based prior effectively bridges the gap between biological responses and chemical structures, enabling the generation of novel molecules that can induce desired gene expression changes.

### 3.5 GENERATION OF DRUG CANDIDATES

After training the prior model, gene expressions corresponding to chemical perturbations can be used for inference to generate drug candidates that might result in similar perturbation effects. The gene expression embeddings $z^{gene}$ are extracted using the pretrained gene expression encoder in the contrastive learning module and subsequently input to the prior model to compute chemical embeddings $z^{chem}$. $z^{chem}$ are then decoded via the SELFIES VAE model to obtain potential drug candidates in the form of novel molecular structures.

## 4 RESULTS AND DISCUSSION

### 4.1 IMPLEMENTATION DETAILS

**Datasets.** The L1000toRNAseq dataset, originally containing 978 landmark genes, was transformed to RNA-seq-like profiles encompassing 23,614 genes using a cycleGAN model as described by Jeon et al. (2022). The dataset includes gene expression profiles from 221 human cell lines treated with over 30,000 chemical and genetic perturbations, resulting in over 3 million expression profiles. We filtered the data for chemical perturbations with 24-hour infection times and 10 μM dosage for the MCF7 cell line, resulting in 3116 genes with high variance (variance $> 0.75$). For

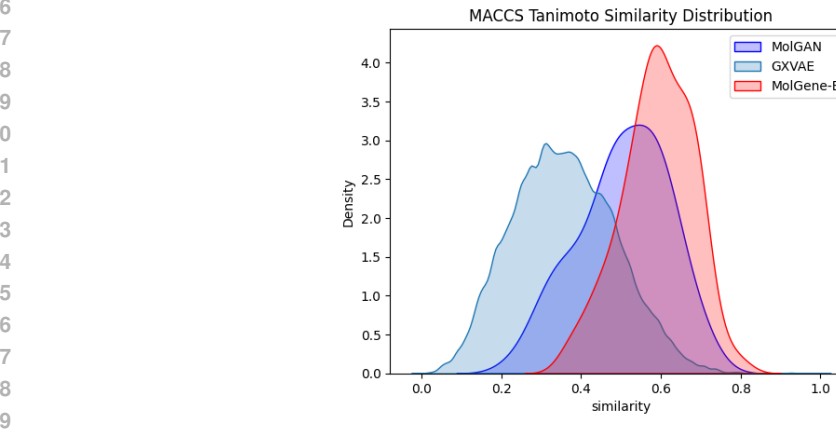

Figure 2: Distributions of MACCS key Tanimoto similarities between molecules generated using gene expression signature induced using CRISPR target knockouts and reference molecules.

training MolGene-E we did a 70-15-15 split to get training, validation, and test sets while ensuring there was no chemical overlap in the data splits.

The Sciplex-3 dataset, sourced from Srivatsan et al. (2020), and harmonized by scPerturb Peidli et al. (2024), includes single-cell transcriptomic profiles of 188 compounds across three cancer cell lines. We focused on the MCF7 cell line, filtering the data to improve quality. The dataset was harmonized using the deep count autoencoder (DCA) method Eraslan et al. (2019) to impute missing values and align with the L1000toRNAseq dataset using an MLP-based network to remove batch effects. This dataset was only used for inference (molecule generation).

**Model Settings.** For the denoising VAE, we used hidden layers of sizes [1024, 512, 256] with layernorm and a dropout rate of 0.3. We used a latent dimension of 128. The weight for KL-term of loss was increased linearly from the first to the last epoch. We trained the model for 400 epochs to achieve good performance on the validation set.

For the SELFIES VAE, it maximizes a lower bound of the likelihood (evidence lower bound (ELBO)) instead of estimating the likelihood directly. We used the pretrained model and architecture identical to the one implemented in MOSES Polykovskiy et al. (2020) to model SELFIES strings. The architecture used a bidirectional Gated Recurrent Unit (GRU) with a linear output layer as an encoder. The decoder was a 3-layer GRU of 512 hidden dimensions with intermediate dropout layers and a dropout rate of 0.2. Training was done with a batch size of 128, utilizing a gradient clipping of 50, KL-term weight linearly increased from 0 to 1 during training. We optimized the model using Adam optimizer with a learning rate of 3e-4.

For the prior model, it is a feed forward MLP architecture with hidden sizes [1024, 512, 256] and a latent dimension of size 128. The model was trained to minimize a mean squared error loss for the reconstruction of chemical embedding space utilizing the aligned spaces from both modalities. The model used a learning rate of 1e-3 and a batch size of 128.

For the training of MolGene-E, we minimize the contrastive learning objective $\mathcal{L}$ (Equation 3) with an approach similar to Radford et al. (2021). MolGene-E was trained for 600 epochs with a batch size of 128 and a learning rate of 1e-4. A projection network with MLP hidden layers [128, 128] was further added to the gene expression encoder. When generating drug candidates, 400 unique chemical candidates are generated by sampling from the latent space for each gene expression profile.

For further details on the training (Algorithm 1,2) and inference (Algorithm 3) process, please refer to Appendix A.

Table 1: Evaluation metrics on CRISPR datset. We mark the best results in bold and the second-best results with underline.

| Model | Validity ↑ | Uniqueness ↑ | Novelty ↑ | Diversity ↑ | SA ↓ |
|---|---|---|---|---|---|
| MolGAN | 0.46 | 0.89 | **1.00** | **1.00** | 4.14 |
| GxVAE | 0.93 | **0.91** | 0.21 | 0.73 | **2.87** |
| MolGene-E | **1.00** | 0.89 | **1.00** | 0.99 | 3.20 |

Table 2: Evaluation metrics on single cell dataset Sciplex-3. We mark the best results in bold and the second-best results with underline.

| Model | Validity↑ | Uniqueness↑ | Novelty↑ | Diversity↑ | SA↓ |
|---|---|---|---|---|---|
| MolGAN | 0.46 | **0.89** | **1.00** | **1.00** | 4.14 |
| GxVAE | 0.42 | 0.23 | 0.98 | 0.81 | **2.85** |
| MolGene-E | **1.00** | **0.89** | **1.00** | 0.99 | 3.20 |

**Evaluation Metrics.** For performance evaluation, the following measures were used. **Tanimoto similarity scores** between reference and generated molecules are computed on encoding molecules to MACCS keys. Generated molecules with higher Tanimoto similarity scores are considered as more potential drug candidates. **Novelty** is the fraction of generated molecules having $\leq 0.4$ tanimoto similarity with the reference molecules. **Uniqueness** is the fraction of distinct molecules generated for each input gene expression profile. The mean of uniqueness for all generated molecules corresponding to their reference molecules was reported. **Validity** and **SA** (synthesizability and accessibility scores) are computed using the RDKit library Riniker & Landrum (2013).

## 4.2 MolGene-E Improves the Success Rate of Inverse Molecule Design

We use a challenge task to evaluate the performance of the molecular generation from gene expressions. If a drug can correctly revert gene expressions from a disease state to a healthy state, the drug could interact with disease-causing genes, i.e., drug targets. In other words, the gene expression changes that are caused by the target gene knock-out or knock-down should be similar to those that result from the chemical perturbation targeting the knock-out/down gene. In our experiments, reference molecules from the test-split of L1000toRNAseq dataset were considered which had single target knock-outs in the CRISPR gene perturbation dataset for the MCF7 cell line. Gene expression profiles for these targets were used for the inference of drug candidates. As shown in Figure 2, MolGene-E outperforms both baseline models, MolGAN and GxVAE, in terms of average Tanimoto similarities computed using the MACCS keys between the generated and reference molecules. For further evaluation of quality of generated molecules, other metrics such as uniqueness, validity, novelty, diversity and synthesizability (SA) were used as listed in Table 1. MolGene-E performs the best overall by consistently achieving higher scores in validity and novelty while maintaining a high level of uniqueness, diversity, and SA. This indicates that MolGene-E not only generates molecules that closely resemble the reference compounds in terms of structural similarity but also proposes novel and diverse chemical scaffolds that are synthetically feasible.

## 4.3 MolGene-E Can Be Applied to Single-Cell Data

For single-cell RNA-seq (scRNA-seq) data we used Sciplex-3 dataset which uses "nuclear hashing" to quantify global transcriptional responses to thousands of independent perturbations at single-cell resolution. The scRNA-seq data as such is extremely sparse, making it hard to use directly with deep learning models. Hence in order to impute missing values we used Deep Count Autoencoder Eraslan et al. (2019). Additionally, to integrate the scRNA-seq dataset with the L1000RNAtoseq dataset, we used an MLP-based neural network model to map gene expression profiles of Sciplex-3 to L1000toRNAseq to introduce homogeneity in the dataset and remove batch effects, as shown in Figure 3. Furthermore, since each chemical sample has several replicates, gene expression profiles

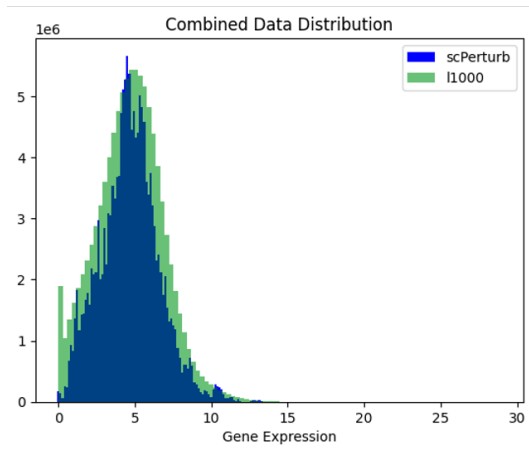

Figure 3: Mean gene expression signatures after harmonizing single cell dataset with L1000.

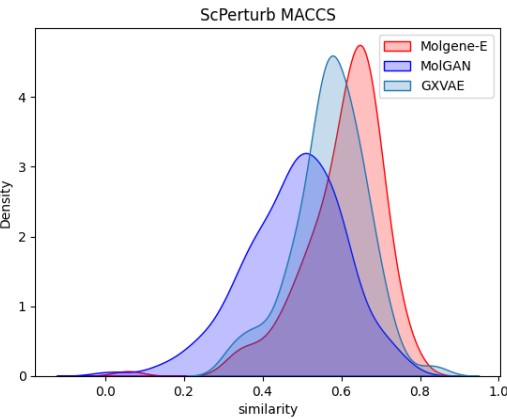

Figure 4: Distributions of MACCS key Tanimoto similarities between reference molecules and molecules generated using gene expression signatures from the Sciplex-3 dataset.

for each chemical perturbation were randomly sampled, and 200 molecules were generated for each gene expression signature. The one with the highest score was chosen as the candidate. Figure 4 shows the Tanimoto similarity distributions of molecules generated using gene expression profiles from the Sciplex-3 single-cell dataset. Results listed in Table 2 indicate that MolGene-E performs the best overall compared to baselines when generating molecules using gene expression profiles for single dataset Sciplex-3. A sample of generated molecules using gene expression perturbation profiles from single-cell data and corresponding reference molecules are presented in Figure 5 in Appendix B.

## 5 CONCLUSION

In this paper, we developed a deep generative model that utilizes phenotypic properties from single-cell omics data to generate high-quality lead candidates for drug discovery. MolGene-E consistently outperforms baseline methods in generating high-quality, hit-like molecules from gene expression profiles obtained from single-cell datasets and gene expressions induced by CRISPR-based knockout targets. This superior performance is demonstrated across *de novo* molecule generation metrics, including novelty, diversity, uniqueness, and synthesizability.

Future work includes incorporating multiple cell lines and conditioning drugs on multi-omics data, leading to a robust framework capable of more accurately reflecting the complex biological environments found *in vivo*. Additionally, expanding the model to integrate diverse datasets will enhance its ability to generalize across different biological contexts, thereby improving its predictive power and utility in identifying effective therapeutic compounds. This approach will pave the way for more personalized and precise drug discovery, ultimately accelerating the development of new treatments and improving patient outcomes.

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

## A  ALGORITHMS FOR TRAINING AND INFERENCE

---

**Algorithm 1** Training the Contrastive Learning Module

---

**Require:** $\mathcal{D}$: Dataset of pairs of SELFIES strings and corresponding gene expressions $(s, g)$, $S$: pre-trained and frozen SELFIES VAE, $G_\theta$: Gene expression VAE
**Require:** $\mathcal{B}$: Set of mini-batches
 1: **for** each mini-batch $\mathcal{M}$ in $\mathcal{B}$ **do**
 2:    **for** each pair $(s, g) \in \mathcal{M}$ **do**
 3:       $z^{chem} \leftarrow S.Encoder(s)$ {Encode SELFIES string}
 4:       $z^{gene} \leftarrow G_\theta.Encoder(g)$ {Encode gene expression}
 5:    **end for**
 6:    Compute average loss $\bar{\ell} \leftarrow \mathcal{L}(z^{chem}, z^{gene})$ over $\mathcal{M}$ (Equation 3)
 7:    Update weights of $G_\theta.Encoder$ using gradient descent with $\bar{\ell}$
 8: **end for**

---

**Algorithm 2** Training the Prior Module

---

**Require:** $\mathcal{D}$: Dataset of pairs of SELFIES strings and corresponding gene expressions $(s, g)$, $S$: pre-trained and frozen SELFIES VAE, $G_\theta$: pre-trained and frozen gene expression VAE via Algorithm 1
**Require:** $\mathcal{B}$: Set of mini-batches
**Require:** Initialize Prior Model $\theta$
 1: **for** each mini-batch $\mathcal{M}$ (batch size=$N$) in $\mathcal{B}$ **do**
 2:    **for** each pair $(s, g) \in \{(s_k, g_k), k \in [1, N]\}$ **do**
 3:       $z^{chem} \leftarrow S.Encoder(s)$ {Encode SELFIES string}
 4:       $z^{gene} \leftarrow G_\theta.Encoder(g)$ {Encode gene expression}
 5:       $\hat{z}^{chem} \leftarrow \theta(z^{gene})$ {Map gene expression to chemical space}
 6:       $\ell \leftarrow \mathcal{L}_{\text{RMSE}}(z^{chem}, \hat{z}^{chem})$ {Compute RMSE loss}
 7:    **end for**
 8:    Compute average loss $\bar{\ell}$ over the mini-batch
 9:    Update weights of Prior Model $\theta$ using gradient descent with $\bar{\ell}$
10: **end for**

---

**Algorithm 3** Inference

---

**Require:** Gene expression $g$
**Require:** $G_\theta$: Gene expression VAE, $S$: SELFIES VAE, $P$: pre-trained and frozen prior model via Algorithm 2
 1: $z^{gene} \leftarrow G_\theta.Encoder(\text{g})$ {Encode gene expression to gene embedding}
 2: $z^{chem} \leftarrow P(z^{gene})$ {Generate chemical embedding using Prior model}
 3: $s_{\text{molecule}} \leftarrow S.Decoder(z^{chem})$ {Decode chemical embedding}
 4: **return** $s_{\text{molecule}}$ {Return molecule represented as SELFIES string}

---

## B  GENERATED MOLECULES FROM SINGLE CELL DATA SCIPLEX-3

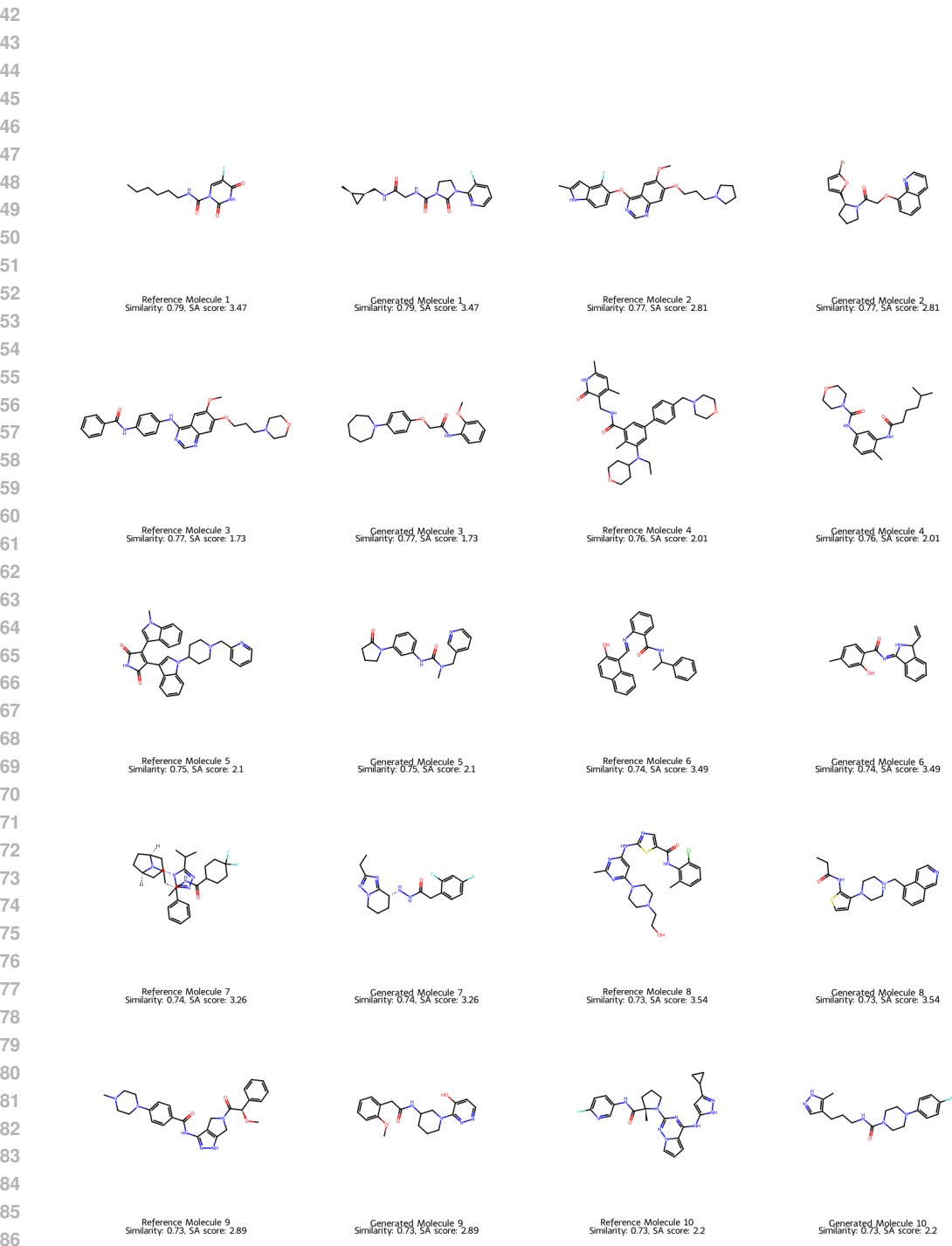

Figure 5: Reference and generated molecules using gene expression profiles from sciplex-3.

