# OpenReview forum: "MolGene-E: Inverse Molecular Design to Modulate Single Cell Transcriptomics"
_ICLR.cc/2025/Conference — ICLR 2025 Conference Withdrawn Submission_

### Official Review · Reviewer_xh78 · 2024-10-23

**Soundness:** 3
**Presentation:** 3
**Contribution:** 4
**Rating:** 8
**Confidence:** 4

**Summary:**

This paper proposes a new task and an interesting method to generate molecular from gene expression profiles, which might enhance the area of molecular design.

**Strengths:**

The task is interesting, and the method is easy to understand. The paper is well-written.

**Weaknesses:**

This paper is interesting and the work is meaningful. However, I still have some questions and concerns, which I intend to discuss with the authors.

1. I am a bit confused about the metric setting. Since the authors have testing datasets for molecules, will it be easier to understand and compare by measuring the similarity of generated molecular and true molecule based on jaccard distance between two molecules? I recommend the authors considering this metric for better understanding.

2. It seems that the authors split the training, validation and testing based on a single study, which might weaken the generalization ability of their model. I would recommend considering this open-problems in single-cell dataset [1] and perform validation by cross-studies settings, which is more sound.

3. It seems that the authors do not report parameter-tuning steps and ablation tests. To ensure the reproductivity, I recommend including such information. For ablation test, it might be helpful to fine-tune the frozen encoder to test if it can improve the model performance.

4. One concern in single-cell data analysis is that the noise level in each cell. Did the authors check if the cells with the same perturbation can give the same reconstruction molecule results? If not, what is the plan of authors to correct them?

[1] https://www.kaggle.com/competitions/open-problems-single-cell-perturbations

**Questions:**

Please see the weakness section.

---

### Official Review · Reviewer_GMMy · 2024-11-01

**Soundness:** 2
**Presentation:** 3
**Contribution:** 3
**Rating:** 5
**Confidence:** 4

**Summary:**

This paper introduces MolGene-E, a deep generative framework for designing molecules conditioned on expression profiles, leveraging a CLIP-like contrastive learning method to align expression and molecular representations. The model demonstrates superior performance on CRISPR and single-cell datasets, positioning itself as a novel tool for inverse molecule design that could significantly benefit drug discovery.

**Strengths:**

The paper is structured well, with clear and informative figures aiding in the understanding of the methodology.
The innovative application of contrastive learning for inverse molecule design offers a fresh approach, capitalizing on the SELFIES VAE model to robustly represent molecular structures.
The model shows strong performance on the MCF7 cancer cell line, surpassing baseline methods in predictive accuracy and molecule generation quality.

**Weaknesses:**

All experiments are conducted on a single cell line (MCF7), raising questions about the method's generalizability across diverse biological settings.
The model requires retraining for each cell line and is constrained by its reliance on paired data, limiting its broader applicability in various disease contexts.
The design does not address the intrinsic noise and sparsity of single-cell data but depends on preprocessing for data refinement. Ablation studies would be helpful to determine the impact of the imputation and alignment steps.

**Questions:**

1. Can the authors provide additional detail on the MLP-based network in Line 242, specifically how it integrates the Sciplex-3 dataset after imputation? What is the training process for this network?
2. For the L1000toRNA dataset split (70-15-15), was this split done randomly, or based on structural similarity? How well does MolGene-E generalize when generating molecules that are structurally distinct from the training set?
3. Is there any overlap between the molecules in the L1000toRNA training set and those in Sciplex-3? If so, how was this controlled for in training and evaluation?
4. How does MolGene-E handle cases where gene expression profiles exhibit high variability across replicates? Is the current model robust to this?
5. More ablation studies would strengthen the evaluation, for instance, in assessing how effectively the prior model maps gene expressions to chemical embeddings. A straightforward comparison could involve using the k-nearest neighbors (KNN) search within the CLIP framework to identify the nearest cross-modality instances. How does this approach compare with the prior model in terms of accuracy and efficiency?
6. How does the model perform on other cell lines?

---

### Official Review · Reviewer_KSSA · 2024-11-03

**Soundness:** 2
**Presentation:** 3
**Contribution:** 3
**Rating:** 3
**Confidence:** 3

**Summary:**

MolGene-E introduces a novel deep generative framework for designing drugs using single-cell transcriptomics data for phenotype-based drug discovery. The authors present a dual-component system that combines a harmonization model for chemical-perturbed transcriptomics data with a contrastive learning-based molecule generator. They benchmark against existing approaches like MolGAN and GxVAE. They conceive a novel task where they take the gene expression of a perturbed cell and try to find a drug that would restore the gene expression back to a healthy cell state.

**Strengths:**

- Addresses a limitation of one-drug-one-target paradigm by leveraging comprehensive gene expression profiles rather than singular targets
- Shows quantitative improvements over baselines (MolGAN, GxVAE) in terms of molecular validity and some generation metrics
- Paper is clearly written
- Formulating the problem as an inverse problem where the target is the unperturbed state is a problem formulation that seems interesting and relevant.

**Weaknesses:**

My main qualm with this paper is that the architecture complexity appears unjustified:
- The method combines multiple components (denoising VAE, SELFIES VAE, contrastive learning module, prior model) without demonstrating the necessity of each
- No ablation studies to show whether all components contribute meaningfully to performance
- The harmonization process between L1000toRNAseq and Sciplex-3 datasets adds even more complexity through cycleGAN and DCA methods

The validation scope is fairly limited:
- All experiments conducted only on MCF7 cell line without justification
- No demonstration of generalizability to other cell types or conditions
- The L1000toRNAseq dataset filtering (24-hour infection times, 10 µM dosage) seems arbitrary
I'm also not that convinced by the metrics. Obviously using SELFIES will lead to perfect validity. Novelity and Uniqueness don't seem as relevant (it is presumably fast to just sample more molecules, and discard duplicates)

Something seems off with Tables 1 and 2 where they show the exact same results for MolGenE. This is either a data entry issue or demands further investigation.

**Questions:**

- Can you explain the reasoning behind restricting the evaluated datasets, e.g. the choise of MCF7 cell line?
- What happens if you use simpler alternatives for the harmonization between L1000toRNAseq and Sciplex-3 datasets?
- How come Tables 1 and 2 show identical metrics for MolGene-E across different datasets?

---

### Official Review · Reviewer_5C1v · 2024-11-04

**Soundness:** 1
**Presentation:** 2
**Contribution:** 1
**Rating:** 1
**Confidence:** 4

**Summary:**

The authors propose to address the problem of inverse molecule design by introducing a new method, MolGene-E, that
1. learns a denoising encoder for gene expression profiles induced by molecules by reconstructing the mean profile for the molecule across replicates
1. learns a SELFIES encoder-decoder
1. fine-tunes the expression profile encoder to align profile embeddings with corresponding compound embeddings using CLIP
1. learns to map profile embeddings to corresponding compound embeddings from the frozen encoders.

The authors show a possibly significant increase in the similarity of generated molecules to the original query molecules, as well as generally improved validity, uniqueness, novelty, diversity and synthesizability/accessibility score performance.

**Strengths:**

- MolGene-E might generate molecules that are more similar to the ground-truth molecules in the test set.
- The choice of SELFIES guarantees the generated molecules are valid.

**Weaknesses:**

- The paper feels very "underdone":
  - The differences in tanimoto similarity distributions shown in Figs 2 and 4 should have been quantified and tested for significance.
  - There should be ablation studies for at least two aspects of the method: the pretraining of a profile VAE (this could have been learned directly during CLIP) and the use of the prior MLP to map $z^{gene} \rightarrow z^{chem}$ (decoding could have been done directly from $z^{gene}$).
  - The datasets used in this paper are highly processed using trained models that could introduce substantial bias into the results. Studies to understand the impact of these choices would have strengthened this paper.

  - The paper claims a number of times that the generated molecules are "hit-like" but I don't see that justified anywhere.
- nitpicks
  - add parentheses around your inline references
  - new relevant paper that should be cited [1]

[1] Fradkin, Philip, et al. "How Molecules Impact Cells: Unlocking Contrastive PhenoMolecular Retrieval." arXiv preprint arXiv:2409.08302 (2024).

**Questions:**

- Were the results for MolGAN and GxVAE obtained from models retrained on the same training set as described in this paper for MolGene-E? It's not clear from the paper.
- It's surprising that so many numbers are $\textit{exactly the same}$ in Tables 1 and 2. Is there an obvious reason for this?
- Why bother with learning a gene encoder prior to the CLIP task? Learning to align to the corresponding molecule embedding would also denoise the expression profiles.
- Why introduce the MLP $z^{gene} \rightarrow z^{chem}$? Isn't the point of the CLIP alignment to ensure that $z^{gene} \approx z^{chem}$ for gene expression profiles induced by the corresponding molecule?

---

### Official Review · Reviewer_SMHM · 2024-11-05

**Soundness:** 3
**Presentation:** 3
**Contribution:** 2
**Rating:** 3
**Confidence:** 5

**Summary:**

In this manuscript, the authors developed a new deep generative framework named MolGene-E to predict new drug candidates based on gene expression profile. The methods were originally proposed and trained on bulk cells and could be adapted to process single cell data. The authors were able to show their methods consistently outperforms baseline methods in
generating high-quality, hit-like molecules from gene expression profiles obtained
from single-cell datasets and gene expressions induced by knocking out targets.

**Strengths:**

The questions proposed in this paper have a certain biological and pharmaceutical value.

**Weaknesses:**

Overall, the proposed manuscript has little novelty on the model itself. It did not bring any new technical innovation to this machine learning community. My feeling is that this paper should be submitted to bioinformatics journals such as Plos CB and Briefings in Bioinformatics.

From a biological point of view, my major concern is the validation of the drug designed by the algorithms. The authors show the proposed drug candidates have a certain similarity to the drug used to perturb the cells. This comparison is rather unsatisfactory and rough. The authors state that the predicted molecules should bind a particular target, why not directly dock and predict molecules and the target proteins and show their predicted binding affinity.

Similar to an existing drug is not enough, changing one key heavy atom might change the binding affinity between the small molecule and its target. The authors need to make sure the translation from the gene expression profile to the small molecule actually learns to generate the useful core molecular regions that are crucial to the drug-target binding.

Another question is the train-test split of the dataset. Many genes/proteins are functional and deleting them might result in the same gene expression profile. For instance, if two proteins A and B form a complex, knocking out protein A or B yields the same result, the complex A-B disappears. The authors should consider ensuring that A and B do not appear with one in the training set and the other in the test set.

**Questions:**

For this community, quite a few biological terms need to explained, such as 'nuclear hashing', the difference between single cell and bulk cell.

---

### Note · Authors · 2024-11-25

I have read and agree with the venue's withdrawal policy on behalf of myself and my co-authors.